# Effects of non-pharmacological interventions on cognitive function in patients with type 2 diabetes mellitus and mild cognitive impairment: A network meta-analysis

Wenzhuo An[1☉], Dongqing Guo[1], Jie Wang[1☉], Xin Chu [iD][2]*

**1** School of Nursing, Chengdu University of Traditional Chinese Medicine, Chengdu, China, **2** Hospital of Chengdu University of Traditional Chinese Medicine, Chengdu, China

☉ These authors contributed equally to this work.
* 18981883803@163.com

## Abstract

### Objective

Non-pharmacological intervention represents a significant therapeutic modality for the cognitive function intervention management of patients with type 2 diabetes accompanied by mild cognitive impairment. However, it remains unclear which intervention measure is the most effective. The objective of this study is to compare and rank the influences of various non-pharmacological interventions on the cognitive function of patients with type 2 diabetes and mild cognitive impairment.

### Methods

Eight databases from the establishment of the database to November 2024 were retrieved. The quality of the literature was evaluated using the RoB2.0 tool. Paired Meta-analysis was conducted using Stata/SE 15.1 software, and Network Meta-analysis was performed using R 4.3.1 software.

### Results

A total of 25 literatures were incorporated, encompassing 5 intervention measures, with a sample size of 2,446 cases. The results indicated that in the pairwise meta-analysis, when the MoCA score was used as the outcome indicator, cognitive training [$MD = 2.3$, 95% $CI$ (1.64, 2.96), $P < 0.01$], exercise therapy [$MD = 2.11$, 95% $CI$ (1.30, 2.92), $P < 0.01$], TCM therapy [$MD = 2.28$, 95% $CI$ (0.76, 3.81), $P < 0.01$], and comprehensive intervention [$MD = 1.98$, 95% $CI$ (1.53, 2.48), $P < 0.01$] were more effective in improving the cognitive status of patients than the control group; when the MMSE was used as the outcome indicator, cognitive training [$MD = 2.03$ 95% $CI$ (1.57, 2.49), $P < 0.01$], exercise therapy [$MD = 2.78$, 95% $CI$ (1.48, 4.08), $P < 0.01$],

**Data availability statement:** All relevant data are within the paper and its Supporting information files.

**Funding:** The author(s) received no specific funding for this work.

**Competing interests:** The authors have declared that no competing interests exist.

and TCM therapy [$MD = 2.09$, 95% $CI$ (1.46, 2.73), $P < 0.01$] were more effective in improving the cognitive status of patients than the control group. The SUCRA ranking revealed that in terms of improving the MoCA scores, the comprehensive intervention (SUCRA 76.9%), cognitive training (SUCRA 63.3%), and TCM therapy (SUCRA 57.9%) were the top 3 preferred treatment measures; for improving the MMSE scores, exercise therapy (SUCRA 78.0%) and cognitive training (SUCRA 73.8%) were the preferred treatment measures.

## Conclusion

The current evidence indicates that cognitive training, exercise therapy, and TCM therapy might be relatively effective intervention approaches for improving the cognitive function of patients with type 2 diabetes mellitus (T2DM) accompanied by mild cognitive impairment (MCI), and cognitive training could potentially be the most efficacious non-pharmacological treatment method. Constrained by the quantity and quality of the studies, more high-quality research is still required in the future to further validate this conclusion.

## 1. Introduction

Diabetes is a group of metabolic disorders characterized by definite hyperglycemia, classified as type 1 diabetes mellitus (T1DM), type 2 diabetes mellitus (T2DM), specific types of diabetes mellitus, and gestational diabetes mellitus [1]. As of the end of 2021, the number of adults with diabetes worldwide has reached 537 million (10.5%), and it is projected to reach 783 million (12.2%) by 2045 [2]. T2DM constitutes 90–95% of the total number of diabetes cases [3]. It is a disease characterized by the progressive insufficiency of insulin secretion by non-autoimmune pancreatic β cells [4], resulting in a persistent hyperglycemic state in patients. Chronic and prolonged hyperglycemia can impair the brain function of patients and cause peripheral vascular complications, thereby leading to cognitive dysfunction in T2DM patients [5–7]. A longitudinal study has confirmed that diabetes increases the risk of mild cognitive impairment (MCI) in adults, and T2DM patients are more prone to cognitive decline, at a rate twice that of normal aging [8].

Mild cognitive impairment (MCI) refers to a progressive deterioration of cognitive function that does not affect the ability of daily living activities and does not meet the diagnostic criteria for dementia [9].A meta-analysis showed that the combined prevalence of MCI in T2DM patients was about 45.0% [10]. The impairments of cognitive functions such as memory and executive function caused by MCI not only exert a negative influence on the self-management ability of patients with diabetes [11], resulting in poor glycemic control [12], but also increase the mortality rate of T2DM patients [13]. MCI is the transitional stage between normal cognitive aging and dementia [14], and it is also the major stage of cognitive function decline and the window period for optimal intervention prior to dementia [15]. The early detection

and timely intervention of MCI can not only enhance the quality of life and prognosis of T2DM patients [16], but also lower medical costs and alleviate the burden on patients' families and society.

In recent years, as numerous studies have confirmed the relationship between type 2 diabetes mellitus (T2DM) and cognitive impairment, patients with T2DM accompanied by mild cognitive impairment (MCI) have drawn extensive attention within the medical field. Owing to the significant individual variations in drug treatment, short therapeutic efficacy, and severe side effects such as stroke and death [17], non-pharmacological interventions have gained a crucial position. Currently, meta-analyses have verified that non-pharmacological intervention measures such as repetitive transcranial stimulation, exercise, virtual reality technology, and acupuncture therapy can enhance the cognitive function of elderly patients with MCI [18]. Nevertheless, this study did not cover patients with T2DM and MCI, and it remains undetermined whether non-pharmacological therapies can improve the cognitive function of adult patients with T2DM and MCI. Hence, this research systematically analyzes the existing evidence from randomized controlled trials to clarify whether various existing non-pharmacological therapies are effective for such patients and ranks the various non-pharmacological therapies, providing evidence-based support for clinical work.

## 2. Materials and methods

This study strictly followed the Preferred Reporting Items for Systematic Reviews and Meta-Analyses (PRISMA) guidelines to ensure transparency and reproducibility of the research process (S1 Checklist), and has been registered on PROSPERO (No. CRD42024613284).

### 2.1. Literature sources and search strategy

PubMed, Web of Science, Embase, the Cochrane Library, China National Knowledge Infrastructure (CNKI), VIP, Wanfang Data and Chinese Biomedical Literature Service System (SionMed) database were searched for randomized controlled trials (RCTS) on the effects of non-pharmacological interventions on cognitive function in patients with type 2 diabetes mellitus (T2DM) and mild cognitive impairment (MCI) from the establishment of the database to November 2024. References of included studies and related systematic reviews were also searched to supplement the relevant literature. The search strategy was determined by combining subject headings with free words. The search terms included: Diabetes Mellitus, Type 2, Cognitive Dysfunction, randomized controlled trial, etc. Different databases used different search strategies, and the specific search strategies are shown in the attachment (S1 Table).

### 2.2. Inclusion and exclusion criteria

#### 2.2.1. Inclusion criteria.

(1) Research Subjects: Patients with type 2 diabetes mellitus (T2DM) accompanied by mild cognitive impairment, without restrictions on age or gender. ① For type 2 diabetes, the diagnostic criteria were referred to those formulated by the World Health Organization (WHO) [19], the American Diabetes Association [20], or the Diabetes Society of the Chinese Medical Association [21]; ② Mild cognitive impairment is based on the diagnostic criteria of the National Institute on Aging-Alzheimer's Association (NIA-AA) in 2011 [22], the 2010 Chinese Guidelines for Diagnosis and Treatment of Dementia and Cognitive Impairment [23], and the internationally recognized diagnostic criteria for MCI proposed by Petersen [24].

(2) Intervention measures: both the experimental group and the control group were treated with conventional western medicine for diabetes to control blood glucose and complications, including oral hypoglycemic drugs biguanides, sulfonylureas, and glinides. On this basis, the control group received usual nursing intervention, including life, diet, exercise, medication guidance and health education, etc. The experimental group adopted non-pharmacological

intervention methods based on the control group. The specific definitions were formulated according to the American College of Sports Medicine (ACSM) [25] and previous studies [26,27]:

Cognitive training: It involves a series of exercises guided by standard tasks aimed at enhancing memory, attention, or executive functions.

Exercise therapy: Refers to planned, structured, and repetitive movements to maintain one or more components of physical health.

Traditional Chinese medicine (TCM) therapy: Generally refers to methods applied to or from the body surface for treatment, excluding oral medications.

Comprehensive intervention: Refers to the combination of two or more of the aforementioned non-drug intervention methods.

(3) Outcome indicators: Cognitive function. The overall cognitive function of patients was measured using validated tests, including the Montreal Cognitive Assessment (MoCA) score [28] and the Mini-Mental State Examination (MMSE) score [29]. MoCA and MMSE are the most commonly used scales for assessing mild cognitive impairment, and the higher the score, the better the cognitive function.

(4) Research methods: Randomized controlled trial (RCT).

**2.2.2. Exclusion criteria.** (1) Subjects with other serious diseases; (2) interventions included traditional Chinese medicine decoction and other non-drug therapies; (3) articles without full text or incomplete data were not available; (4) duplicate publications.

## 2.3. Literature screening and data extraction

The literature was managed using Endnote X9 software. Two researchers separately read the title and abstract of the articles for preliminary screening, and then read the full text for further exclusion. In case of disagreement, the two parties could discuss by themselves, or negotiate with a third party researcher. The main contents of data extraction included: (1) basic information of the study (first author, publication time, etc.); (2) the baseline data of the study subjects (gender, age, sample size, etc.); (3) intervention measures, time and frequency of intervention; (4) outcome indicators and data of the study.

## 2.4. Literature quality evaluation

Two researchers used the Cochrane Risk of Bias Assessment Tool 2.0 (RoB2.0) [30] to evaluate the quality of the included literature, covering the following six aspects: randomization process, deviation from the established intervention measures, missing outcome data, outcome measurement, selective reporting of research results, and overall bias. Each dimension was classified as "yes" (low risk), "no" (high risk), or "unclear" (moderate risk). In case of disagreement, they would consult the third researcher to resolve the issue.

## 2.5. Statistical methods

**2.5.1. Pairwise meta-analysis.** A paired Meta-analysis was conducted using Stata/SE 15.1 software (StataCorp LP, College Station, TX, USA) to clarify the effects of various non-pharmacological interventions compared to the control group. The statistical results were expressed using mean difference (*MD*) and 95% confidence interval (*CI*). A value of $|MD| < 0.2$ was considered a minor effect size, $0.2 \le |MD| < 0.5$ was a small effect size, $0.5 \le |MD| < 0.8$ was a moderate effect size, and $|MD| \ge 0.8$ was a large effect size [31].Heterogeneity was evaluated by the statistics $I^2$ and $P$ value. If $P \ge 0.1$ and $I^2 < 50\%$, it indicated low heterogeneity among studies, and a fixed-effects model was adopted; otherwise, a random-effects model was employed. Statistical significance was defined as $P < 0.05$.

**2.5.2. Network meta-analysis.** The gemtc package was invoked in the R 4.3.1 software (R Core Team, 2023) to conduct Bayesian network Meta-analysis. A network evidence map was drawn to reveal the associations among various interventions. Nodes in the evidence network represented the interventions, and straight lines indicated the existence of direct comparison evidence between two interventions. The thicker the straight line, the greater the number of studies that directly compared the two interventions. When a closed loop was formed among the interventions, the node-splitting method was employed to test for inconsistency. If the $P$ value between the direct comparison and the indirect comparison was > 0.05, it was judged as consistent. If no closed loop is formed, a consistency model will be used for the network Meta-analysis. If the overall heterogeneity parameter $I^2$ is ≤ 50%, the fixed-effect model will be adopted; otherwise, the random-effect model will be selected. Sensitivity analysis will be conducted for the outcome indicators with high heterogeneity, and the sources of heterogeneity will be explored through Meta regression analysis. Due to the influence of the number of studies, this study only conducts sensitivity analysis and Meta regression for the included studies with MoCA score as the outcome indicator. The potential scale reduction factor (PSRF) was used to determine the simulation effect of the model. A satisfactory convergence was achieved when the PSRF value was between 1 and 1.05, and the closer to 1, the more stable the result. The surface under the cumulative rank curve (SUCRA) was utilized to present the possibility of each intervention becoming the best intervention. The SUCRA value ranges from 0 to 1. The higher the value, the better the effect of the intervention measure [32]. A funnel plot was drawn using Stata/SE 15.1 software to determine publication bias.

## 3. Results

### 3.1. Literature screening process and results

A total of 2424 relevant literatures were obtained in the initial inspection, and 5 literatures were supplemented by reference tracing. EndnoteX9 software was used to screen the abstracts and titles of the included literatures, and 76 literatures were initially included. After reviewing the full texts and conducting systematic analyses, we excluded 2 non-randomized controlled trials, 9 studies with inconsistent intervention measures, 14 studies with inconsistent outcome indicators, 12 studies with mismatched population characteristics, 13 studies due to unavailability of full texts or incomplete data, and 1 study due to duplicate publication (S2 Table). Ultimately, a total of 25 randomized controlled trials (RCTs) were included in the analysis. The literature screening process is illustrated in Fig 1.

### 3.2. Literature characteristics

A total of 25 literatures [33–57] were included, among which 23 [33–37,39–51,53–57] were double-arm studies and 2 [38,52] were three-arm studies. During the statistical analysis, the three-arm studies were split into two double-arm studies. The intervention measures encompassed 10 items of cognitive training [33,34,37,38,40–43,45,52], 11 items of exercise therapy [36,38,39,48,49,52–57], 2 items of TCM therapy [35,47], 4 items of comprehensive intervention [44,46,50,51], and the usual care in the control group. The basic characteristics of the included literatures are presented in Table 1.

### 3.3. Evaluation of literature quality

Two researchers conducted a rigorous quality assessment of the included literature using the Cochrane Risk of Bias Assessment Tool 2.0 (RoB2.0). A total of 25 literatures [33–57] were included. The results of bias risk assessment indicated that 14 literatures [33,37,39–41,44,45,48,49,51,54–57] reported the specific methods for generating random sequences; 2 literatures [54,55] reported the specific implementation details of allocation concealment; 4 literatures [44,50,54,55] reported blinding; 1 literature [57] reported that the number and reasons for missing data between groups were unbalanced, making it impossible to determine whether there were other biases; the selective reporting of study results was complete in all literatures. See Fig 2 and S3 Table.

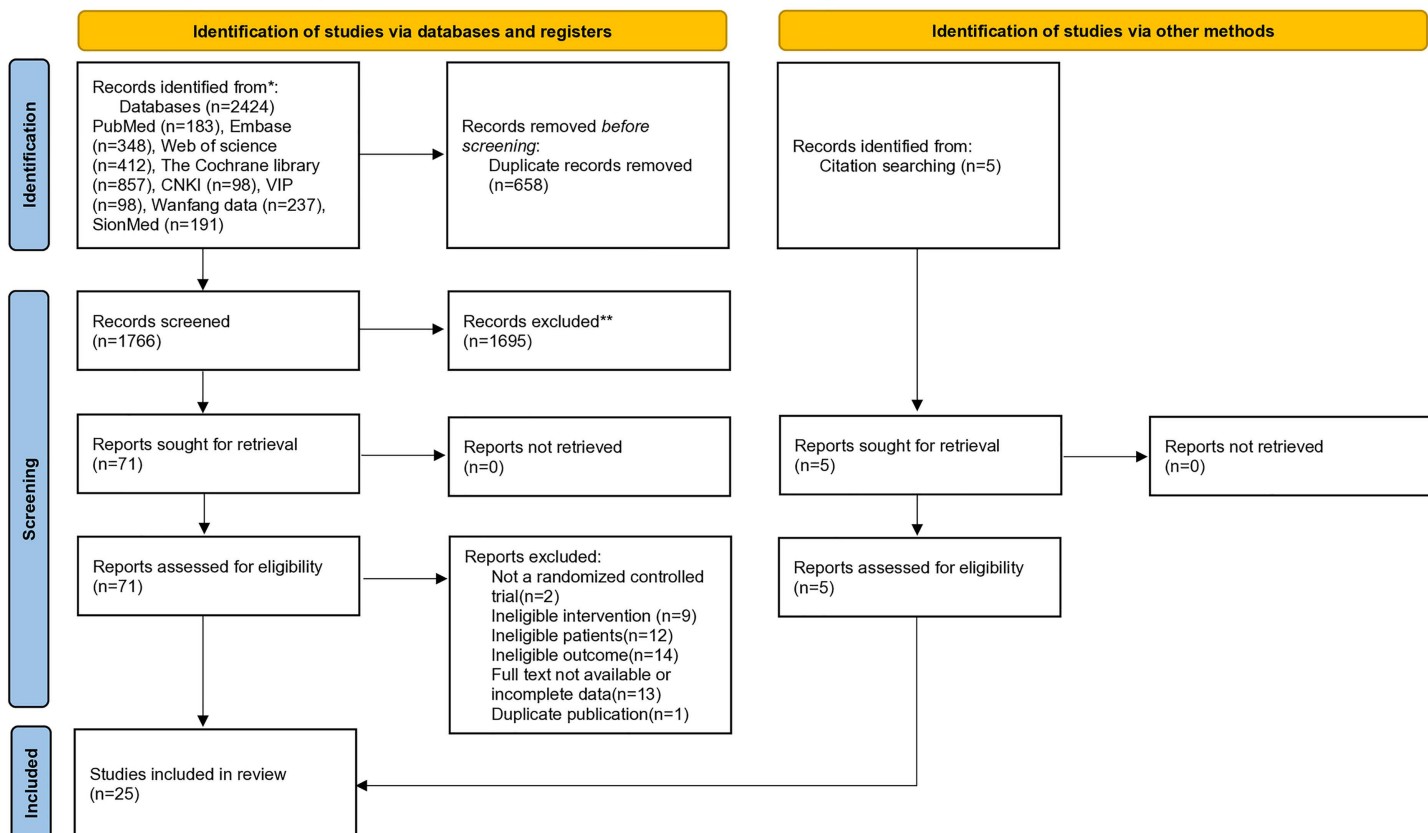

**Fig 1. Flowchart for the selection of studies.**

### 3.4. Results of pairwise meta-analysis

The 23 included studies [33–35,37–40,42–57] adopted the MoCA score, while 11 studies [34–37,39,41,42,46,47,50,57] employed the MMSE score to evaluate the cognitive function of patients. The intervention measures for the two outcome indicators were subjected to pairwise Meta-analysis, and the selection of the effect model was determined based on the heterogeneity. Due to the large heterogeneity among the included studies, a sensitivity analysis was conducted to eliminate individual studies one by one. In the comprehensive intervention where MoCA score was used as the outcome indicator, after eliminating the study by Ye et al. [44], the heterogeneity decreased. The analysis suggests that this might be due to the fact that in the comprehensive intervention measures, the cognitive training component of this study accounted for a relatively large proportion while other intervention measures were relatively less prominent. In the cognitive training with MMSE score as the outcome indicator, after eliminating the study by Liang et al. [34], the heterogeneity decreased. The analysis might be attributed to the relatively short duration of the intervention in this study. There was no significant change in the heterogeneity of other intervention measures. The Meta-analysis results were robust and are shown in S1 Fig. The results indicated that when the MoCA score was used as the outcome indicator, cognitive training [$MD = 2.3$, 95% $CI$ (1.64, 2.96), $P < 0.01$], exercise therapy [$MD = 2.11$, 95% $CI$ (1.30, 2.92), $P < 0.01$], TCM therapy [$MD = 2.28$, 95% $CI$ (0.76, 3.81), $P < 0.01$], and comprehensive intervention [$MD = 1.98$, 95% $CI$ (1.53, 2.48), $P < 0.01$] were more effective in improving the cognitive status of patients compared with the control group; when the MMSE was used as the outcome indicator, cognitive training [$MD = 2.03$, 95% $CI$ (1.57, 2.49), $P < 0.01$], exercise therapy [$MD = 2.78$, 95% $CI$ (1.48, 4.08),

**Table 1. Basic characteristics of included studies.**

| Included literature | Gender (male/female) | Age (T/C, years) | Number of cases (T/C) | Interventions | | Intervention Duration | Frequency intervention | Outcome Measures |
|---|---|---|---|---|---|---|---|---|
| | | | | T | C | | | |
| Lin [33] 2019 | 45/35 | 67.5±3.4/67.4±3.5 | 40/40 | Cognitive training | Usual care | NA | NA | ① |
| Liang [34] 2013 | 30/32 | NA | 31/31 | Cognitive training | Usual care | 1 month | 1 time/day, 1 hour/time | ①② |
| Yan [35] 2019 | NA | NA | 30/30 | TCM therapy | Usual care | 3 months | 6 times/week, 1day/time | ①② |
| Liang [36] 2017 | 52/48 | 62.0±5.2/65.0±6.1 | 50/50 | Exercise therapy | Usual care | 3 months | 2 times/day, 20~30 min/time | ② |
| Wu [37] 2018 | 43/37 | 65.9±5.6/65.6±5.5 | 40/40 | Cognitive training | Usual care | 3 months | 1 time/day, 15 min/time | ①② |
| Zhang [38] 2017 | 50/59 | 67.3±6.9/69.5±8.0/69.8±6.7 | 35/36/38 | Cognitive training/ Exercise therapy | Usual care | 3 months | 3 times/week, 30 min/time; 3 times/week, 40~50 min/time | ① |
| Liu [39] 2020 | 138/118 | 76.8±4.2/77.2±4.0 | 128/128 | Exercise therapy | Usual care | 6 months | 3 times/week, 40 min/time | ①② |
| Xu [40] 2021 | 78/45 | 67.9±5.2/66.7±4.6 | 62/61 | Cognitive training | Usual care | 7 months | 1 time/day, 30~90 min/time | ① |
| Li [41] 2019 | 60/50 | 69.8±3.8/69.8±3.9 | 55/55 | Cognitive training | Usual care | NA | 2 times/day | ② |
| Hu [42] 2015 | 34/45 | 64.3±7.9/66.9±8.0 | 39/40 | Cognitive training | Usual care | 3 months | 6 times/week, 30~60 min/time | ①② |
| Lv [43] 2016 | NA | NA | 41/39 | Cognitive training | Usual care | 6 months | 5 times/week, 1.5hours/time | ① |
| Ye [44] 2020 | 103/97 | 49.9±7.0/47.1±6.2 | 100/100 | Comprehensive intervention | Usual care | NA | 1 time/day, 2 hours/time | ① |
| Zhang [45] 2023 | 35/61 | 69.4±6.7/68.8±5.4 | 48/48 | Cognitive training | Usual care | 6 months | NA | ① |
| Wang [46] 2021 | 44/40 | 65~69/65~69 | 42/42 | Comprehensive intervention | Usual care | 3 months | 2 times/week, 20 min/time; 2 times/day, 15~20 min/time | ①② |
| Li [47] 2022 | 44/40 | 50~80 | 42/42 | TCM therapy | Usual care | 3 months | 3 times/day, 15–20 min/time | ①② |
| Hu [48] 2019 | 49/47 | 65.8±3.6/65.9±3.7 | 49/47 | Exercise therapy | Usual care | 3 months | 4 times/week, 40~60 min/time | ① |
| Wei [49] 2021 | 57/63 | 72.2±6.3/71.8±6.7 | 60/60 | Exercise therapy | Usual care | 3 months | 5 times/week, 40~60 min/time | ① |
| Chen [50] 2019 | 34/26 | 65.3±4.0 | 40/20 | Comprehensive intervention | Usual care | NA | 2 times/week, 40 min/time; 3 times/week, 30 min/time | ①② |
| Feng [51] 2019 | NA | NA | 45/45 | Comprehensive intervention | Usual care | 3 months | 2 times/week, 30 min/time; 5 times/week, 30 min/time; 3 times/week, 30 min/time | ① |
| Matveeva [52] 2019 | 33/57 | 54.3±12.0/58.7±11.5/56.0±11.0 | 30/30/30 | Cognitive training/ Exercise therapy | Usual care | 6 months | 2 times/week, 45 min/time; 2 times/week, 60 min/time | ① |

*(Continued)*

**Table 1.** (Continued)

| Included literature | Gender (male/female) | Age (T/C, years) | Number of cases (T/C) | Interventions T | Interventions C | Intervention Duration | Frequency intervention | Outcome Measures |
|---|---|---|---|---|---|---|---|---|
| Ghodrati [53] 2023 | 0/21 | 58.8±1.5/55.8±1.5 | 12/9 | Exercise therapy | Usual care | 3 months | 3 times/week, 65 min/time | ① |
| Sewillam [54] 2024 | 18/19 | 69.4±4.0/69.7±4.4 | 27/10 | Exercise therapy | Usual care | 3 months | 3 times/week, 60 min/time | ① |
| Chen [55] 2023 | 100/118 | 67.6±5.0/67.6±5.4 | 107/111 | Exercise therapy | Usual care | 6 months | 3 times/week, 1hour/time | ① |
| Zhu [56] 2015 | NA | 69.9±6.4 | 37/41 | Exercise therapy | Usual care | 12 months | 5 times/week, 40 min/time | ① |
| Ploydang [57] 2023 | 12/21 | 68.9±3.7/69.2±5.3 | 16/17 | Exercise therapy | Usual care | 3 months | 3 times/week, 60 min/time | ①② |

Note: T: experimental group; C: control group; NA: not reported; ①=MoCA, ②=MMSE.

$P < 0.01$], and TCM therapy [$MD = 2.09$, 95% $CI$ (1.46, 2.73), $P < 0.01$] were more effective in improving the cognitive status of patients compared with the control group, as presented in Table 2 and S2 and S3 Figs.

### 3.5. Results of network meta-analysis

**3.5.1. Network evidence map between interventions.** The network relationships between the various treatments are shown in Fig 3: letters represent the corresponding interventions, and the thickness of the lines between different letters represent the number of studies.

**3.5.2. MoCA score. 3.5.2.1. Consistency analysis and heterogeneity analysis:** A total of 23 literatures were included for the outcome indicators, involving 5 intervention measures. Among them, there were 9 items of cognitive training, 10 items of exercise therapy, 2 items of TCM therapy, 4 items of comprehensive intervention and 4 items of routine care. There were 21 two-arm trials and 2 three-arm trials, with closed-loop formation. The consistency analysis of the included data was performed using the node-splitting method. The results showed that $P > 0.05$, suggesting that the results of the direct and indirect comparisons of the data within the studies were consistent. In the network meta-analysis, heterogeneity test was conducted, and the results showed that $I^2$. Pair = 82.67, $I^2$.cons = 82.03. The data were analyzed using both the fixed effect model and the random effect model. Under the random effect model, Dbar = 47.46, pD = 42.88, DIC = 90.34, and $I^2 = 1\%$. Under the fixed effect model, Dbar = 152.65, pD = 27.04, DIC = 179.69, and $I^2 = 69\%$. Therefore, the consistency model and the random effect model were selected for analysis.

**3.5.2.2. Cumulative probability ranking:** SUCRA ranking results showed that comprehensive intervention (76.9%)> cognitive training (63.3%)> TCM therapy (57.7%)> exercise therapy (51.7%)> usual care (0.2%). The closer the SUCRA value of the intervention measure is to 1, the greater the probability that it is the optimal intervention measure. See Fig 4 and S4 Table.

**3.5.2.3. Network meta-analysis and convergence analysis:** The results of network Meta-analysis showed that cognitive training [$MD = 2.4$, 95% $CI$ (1.55, 3.27)], exercise therapy [$MD = 2.2$, 95% $CI$ (1.37, 3.04)], TCM therapy [$MD = 2.27$, 95% $CI$ (0.49, 4.04)] and comprehensive intervention [$MD = 2.72$, 95% $CI$ (1.36, 4.04)] were all superior to conventional care, and the difference was statistically significant. And the $MD$ values were all greater than 0.8, indicating a large effect size. The improvement in the patients' cognitive function was quite significant, which was consistent with the results of direct comparisons, see Table 3 and S1 File. The PSRF value of the model after 50 000 iterations was 1.00, suggesting that the convergence was good, indicating that the degree of fitting of the established model was satisfactory.

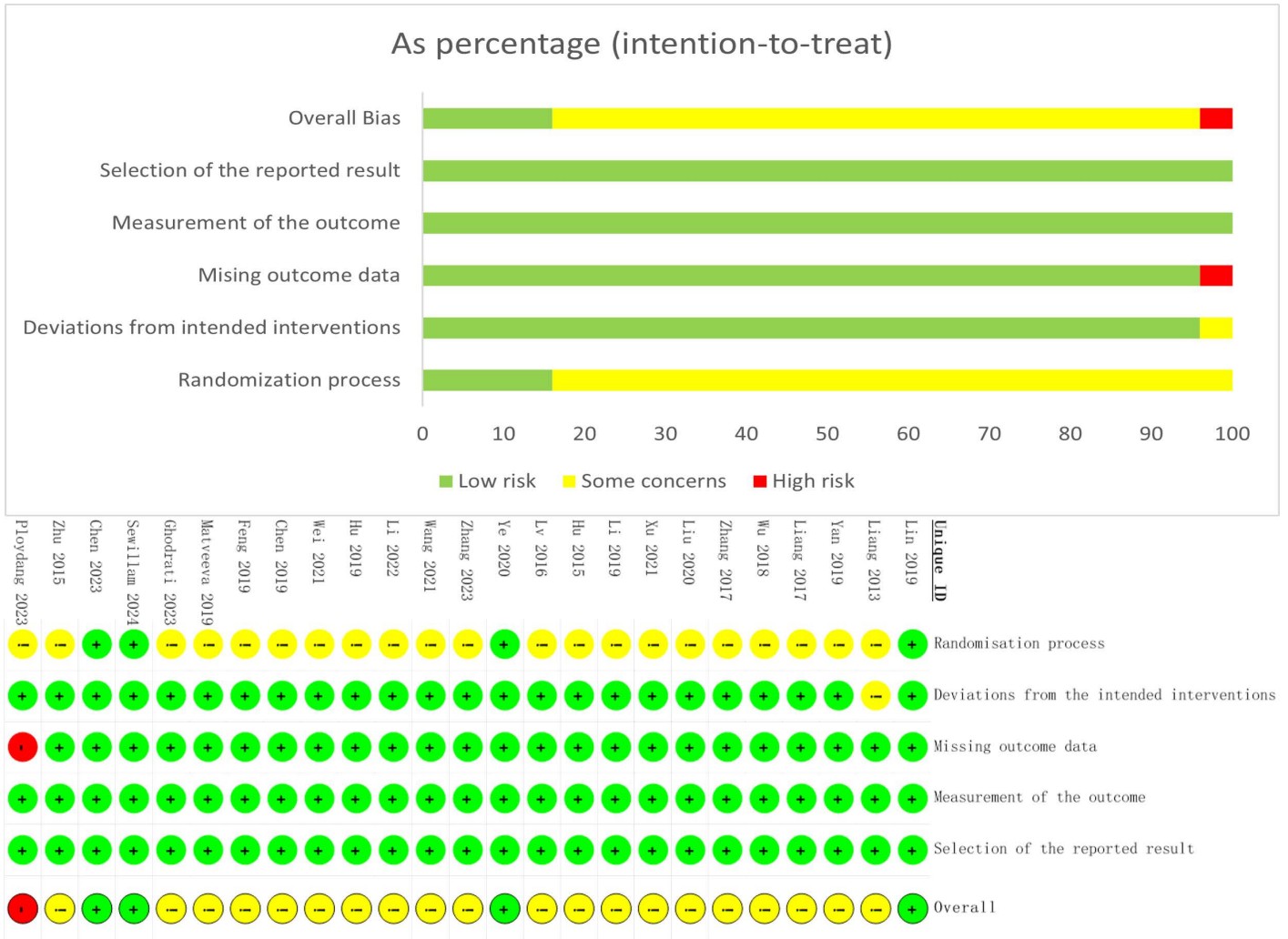

**Fig 2. Quality assessment of the included studies.**

### 3.5.3. MMSE score. 3.5.3.1. Consistency and heterogeneity analysis:
A total of 11 literatures were included for the outcome indicators, involving 5 intervention measures. Among them, there were 4 items of cognitive training, 3 items of exercise therapy, 2 items of TCM therapy, 2 items of comprehensive intervention and 4 items of routine care. There were no direct comparisons among various non-pharmacological therapies, and no closed loop was formed. A consistency model analysis was conducted on the included data. After 50,000 iterations, a heterogeneity test was performed in the network Meta-analysis, resulting in $I^2.Pair = 83.98$, $I^2.cons = 83.96$. The data were analyzed using both the fixed effect model and the random effect model. Under the random effect model, $Dbar = 22.27$, $pD = 21.12$, $DIC = 43.38$, and $I^2 = 6\%$. Under the fixed effect model, $Dbar = 54.78$, $pD = 15.00$, $DIC = 69.78$, and $I^2 = 62\%$. Therefore, the random effect model was selected for analysis.

**3.5.3.2. cumulative probability sorting:** SUCRA ranking results showed that exercise therapy (78.0%)> cognitive training (73.8%)> TCM therapy (60.0%)> comprehensive intervention (33.1%)> usual care (5.1%). The closer the SUCRA value of the intervention measure is to 1, the greater the probability that it is the optimal intervention measure. See Fig 5 and S4 Table.

**Table 2. The results of pairwise meta-analysis.**

| Outcome Measures | Interventions | Number of included studies | Results of heterogeneity test | | Effect model | Meta-analysis Results | |
|---|---|---|---|---|---|---|---|
| | | | P | I2 | | MD (95% CI) | P |
| MoCA | Cognitive training | 9[33, 34, 37, 38, 40, 42, 43, 45, 52] | < 0.01 | 66.5% | Random effects model | 2.3 (1.64, 2.96) | < 0.01 |
| | Exercise therapy | 10[38, 39, 48, 49, 5253545556-57] | < 0.01 | 85.0% | Random effects model | 2.11 (1.30, 2.92) | < 0.01 |
| | TCM therapy | 2[35, 47] | 0.02 | 81.6% | Random effects model | 2.28 (0.76, 3.81) | < 0.01 |
| | Comprehensive intervention | 3[46, 50, 51] | 0.96 | 0% | Fixed effects model | 1.98 (1.53, 2.48) | < 0.01 |
| MMSE | Cognitive training | 3[37, 41, 42] | 0.23 | 32.0% | Fixed effects model | 2.03 (1.57, 2.49) | < 0.01 |
| | Exercise therapy | 3[36, 39, 57] | < 0.01 | 78.8% | Random effects model | 2.78 (1.48, 4.08) | < 0.01 |
| | TCM therapy | 2[35, 47] | 0.82 | 0% | Fixed effects model | 2.09 (1.46, 2.73) | < 0.01 |
| | Comprehensive intervention | 2[46, 50] | < 0.01 | 85.5% | Random effects model | 1.18 (−1.70, 4.07) | 0.42 |

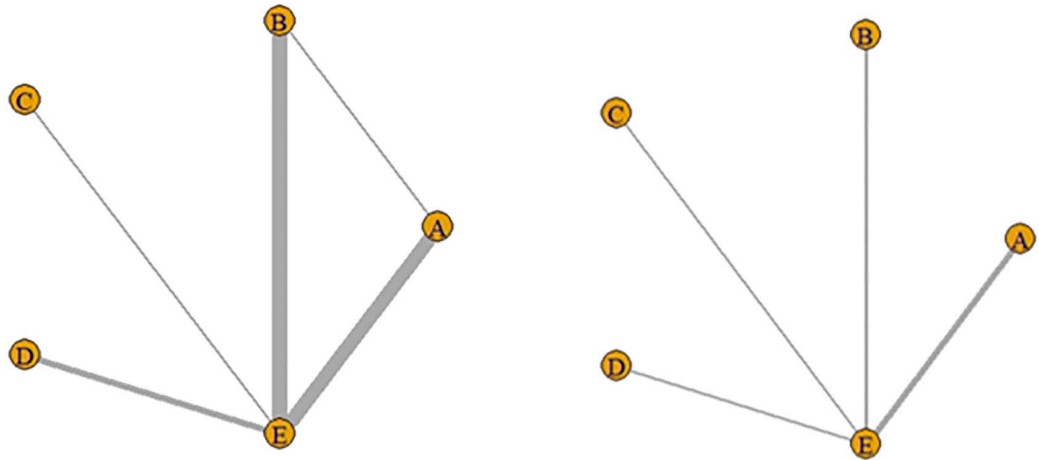

**Fig 3. Network diagram of MoCA (left), MMSE (right).** Notes: A: cognitive training, B: exercise therapy, C: TCM therapy, D: comprehensive intervention, and E: usual care.

**3.5.3.3. Network meta-analysis and convergence analysis:** The results of network meta-analysis showed that cognitive training [$MD = 2.61$, 95% $CI$ (1.01, 4.23)] and exercise therapy [$MD = 2.77$, 95% $CI$ (0.86, 4.69)] were better than usual care, and the difference was statistically significant. And the $MD$ values were all greater than 0.8, indicating a large effect size. The improvement in the patients' cognitive function was quite significant, as shown in Table 4 and S1 File. The PSRF value of the model after 50 000 iterations was 1.00, indicating that the convergence was good, indicating that the degree of fitting of the model was satisfactory

**3.5.4. Sensitivity analysis.** A sensitivity analysis was conducted on the outcome indicators of MoCA. The study with the smallest sample size was excluded for analysis. The results showed that I2.Pair = 82.55, I2.cons = 82.93. The data were analyzed using both the fixed effect model and the random effect model. Under the random effect model, Dbar = 46.10, pD = 41.73, DIC = 87.82, and $I^2 = 2\%$. Under the fixed effect model, Dbar = 151.54, pD = 26.058, DIC = 117.60, and $I^2 = 70\%$. The SUCRA results indicated that comprehensive intervention (76.4%)> cognitive training (63.1%)> traditional Chinese medicine therapy (57.5%)> exercise therapy (52.7%)> routine care (0.2%). This result was not significantly different from the previous result after excluding the study, and the ranking order remained the same. See Appendix S5 Table.

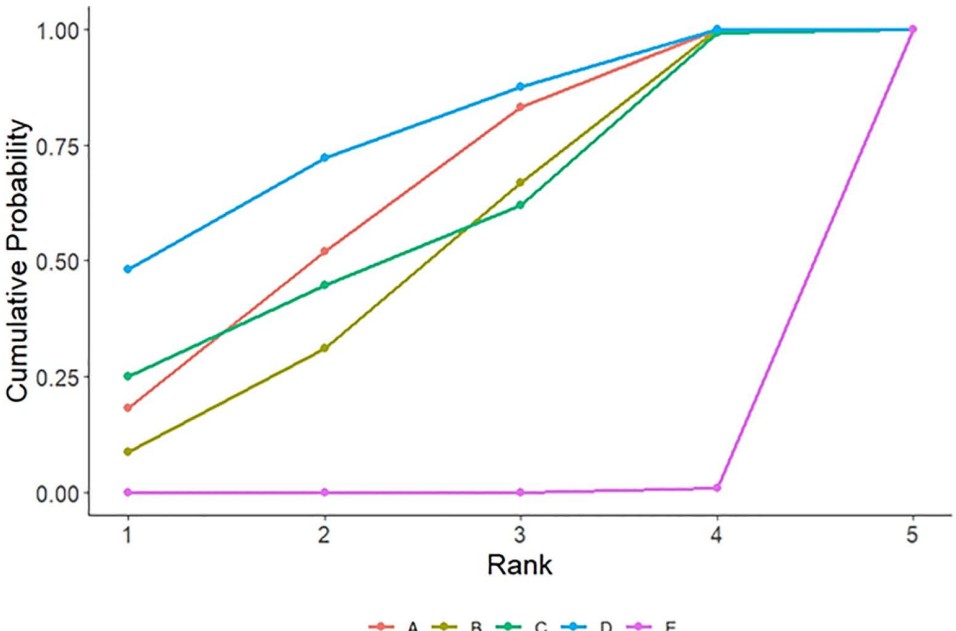

**Fig 4. Cumulative Probability Ranking of MoCA Scores for Different Non-pharmacological Interventions.**

**Table 3. Network meta-analysis results of MoCA score [*MD* (95% *CI*)].**

| | A | B | C | D | E |
|---|---|---|---|---|---|
| A | 0 | | | | |
| B | 0.2 (−0.92, 1.32) | 0 | | | |
| C | 0.12 (−1.83, 2.09) | −0.08 (−2.02, 1.89) | 0 | | |
| D | −0.32 (−1.88, 1.31) | −0.53 (−2.07, 1.09) | −0.45 (−2.65, 1.80) | 0 | |
| E | 2.40 (1.55, 3.26)* | 2.19 (1.36, 3.04)* | 2.28 (0.50, 4.03)* | 2.72 (1.35, 4.03)* | 0 |

Note: A: cognitive training, B: exercise therapy, C: TCM therapy, D: comprehensive intervention, E: conventional nursing; The difference between the control group and the experimental group was statistically significant (*$P<0.05$).

### 3.6. Publication bias analysis

A comparative-corrected funnel plot analysis was conducted for the MoCA outcome indicators. The results indicated that most of the studies were basically distributed on both sides of the funnel plot and were relatively symmetrical, suggesting a relatively small possibility of publication bias, as shown in Fig 6.

### 4. Discuss

Patients with T2DM are more prone to develop MCI [58], and both share similar pathophysiological characteristics, such as insulin resistance, inflammatory stress, and amyloid aggregation, etc. [59]. Compared to non-diabetic patients, the risk of cognitive impairment and dementia in T2DM is approximately 1.5 to 2 times higher [60]. As dementia is irreversible, MCI, an unstable intermediate state, has drawn extensive attention in the medical field. A large number of intervention studies have explored effective ways to improve or delay cognitive decline during the MCI stage in T2DM patients. Among them, non-pharmacological intervention is an important treatment approach for cognitive function intervention and management in T2DM patients with MCI.

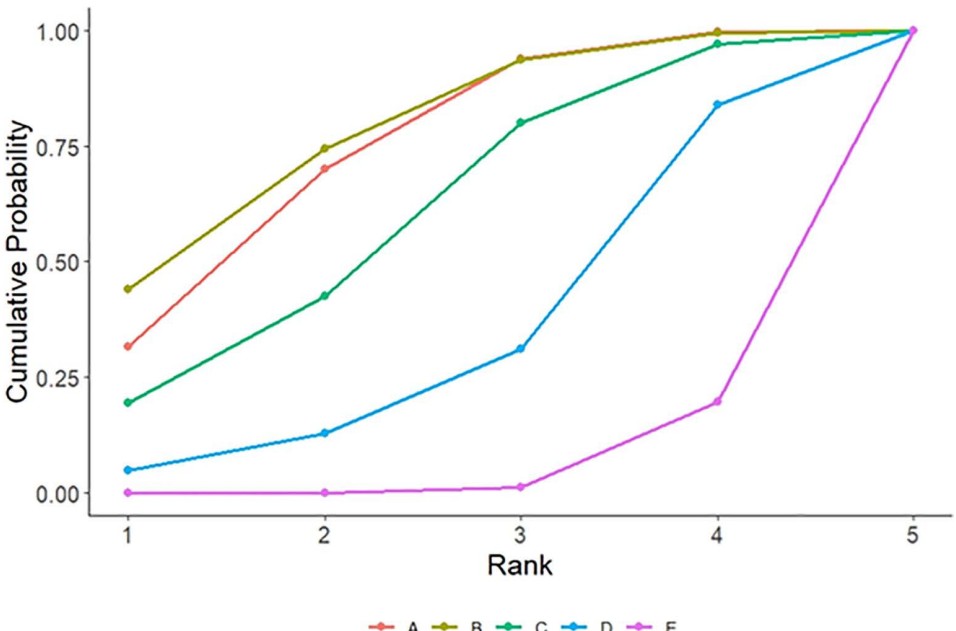

**Fig 5. Cumulative Probability Ranking of MMSE Scores for Different Non-pharmacological Interventions.**

**Table 4. Network meta-analysis results of MMSE score [*MD* (95% *CI*)].**

|   | A | B | C | D | E |
|---|---|---|---|---|---|
| A | 0 | | | | |
| B | −0.17 (−2.68, 2.38) | 0 | | | |
| C | 0.46 (−2.40, 3.31) | 0.63 (−2.44, 3.65) | 0 | | |
| D | 1.55 (−1.46, 4.38) | 1.71 (−1.50, 4.72) | 1.09 (−2.37, 4.39) | 0 | |
| E | 2.61 (1.01, 4.23)* | 2.78 (0.85, 4.70)* | 2.15 (−0.19, 4.51) | 1.06 (−1.29, 3.60) | 0 |

Note: The difference between the control group and the experimental group was statistically significant (*$P < 0.05$)

## 4.1. Main findings

This study encompassed 25 randomized controlled trials (RCTs), involving 2,446 patients and covering 5 intervention measures, summarizing the latest evidence regarding the influence of non-pharmacological interventions on the cognitive function of patients with type 2 diabetes mellitus and mild cognitive impairment. The results indicated that in terms of improving the MoCA score, cognitive training [*MD* = 2.4, 95% *CI* (1.55, 3.27)], exercise therapy [*MD* = 2.2, 95% *CI* (1.37, 3.04)], TCM therapy [*MD* = 2.27, 95% *CI* (0.49, 4.04)] and comprehensive intervention [*MD* = 2.72, 95% *CI* (1.36, 4.04)] were all superior to conventional care. And the *MD* values were all greater than 0.8, indicating a large effect size. The improvement in the patients' cognitive function was quite significant. Comprehensive intervention (SUCRA 76.9%), cognitive training (SUCRA 63.3%), and TCM therapy (SUCRA 57.9%) were the top three preferred treatment measures. Regarding the improvement of MMSE, cognitive training [*MD* = 2.61, 95% *CI* (1.01, 4.23)] and exercise therapy [*MD* = 2.77, 95% *CI* (0.86, 4.69)] were superior to usual care. And the *MD* values were all greater than 0.8, indicating a large effect size. The improvement in the patients' cognitive function was quite significant. Exercise therapy (SUCRA 78.0%) and cognitive training (SUCRA 73.8%) are the better treatment measures. Therefore, by combining the SUCRA ranking, the *MD*

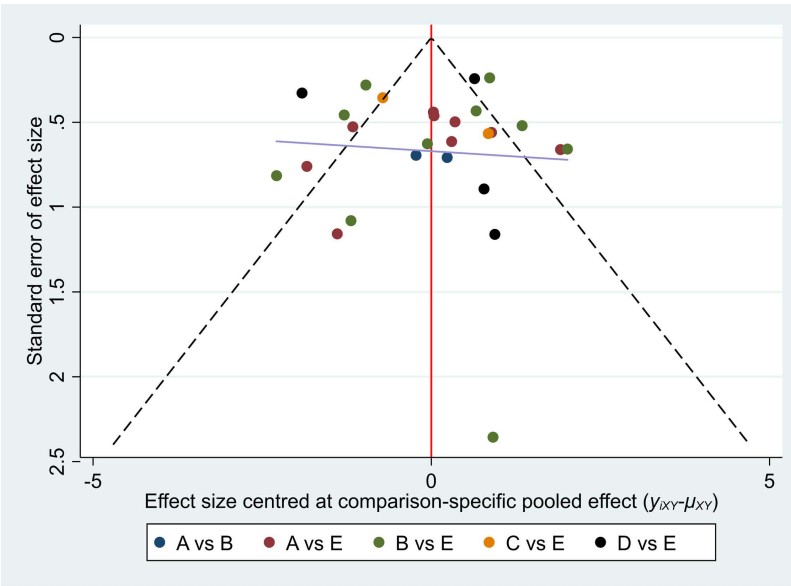

**Fig 6. MoCA bias analysis funnel plot.**

value, and the 95% *CI*, it can be seen that in terms of improving the cognitive function of patients, the effect of cognitive training is the best, and the conclusion is relatively reliable.

### 4.2. Potential mechanisms of non-pharmacological treatments on cognitive function in patients with T2DM and MCI

Cognitive training refers to the use of a series of standardized tasks to provide guided training for specific aspects of cognition (such as memory, attention, language and executive ability, etc.). It is an effective non-pharmacological treatment measure for improving cognitive functions [61]. Numerous studies have likewise confirmed that cognitive training, as a non-invasive, low-cost, and easily implementable therapeutic approach, can bring about immediate cognitive benefits for patients with MCI [62–64]. Additionally, Belleville et al. [65] conducted a five-year follow-up study on elderly individuals with MCI to investigate the lasting impact of cognitive training. The results show that the participants who received cognitive training experienced a smaller decline in delayed memory and maintained their MoCA scores. This indicates that cognitive training can alleviate memory decline and relieve clinical symptoms in elderly individuals with MCI over an extended period of time.

The core mechanism lies in the brain's plasticity, which can be divided into structural plasticity and functional plasticity [66]. Structural plasticity is mainly accomplished through learning and training, causing minute structural alterations in the quantity, quality, and distribution of brain neurons, dendrites, and synapses. Functional plasticity is achieved through learning and training, which brings about changes in the capacity and connections of synapses [67]. This provides a robust theoretical foundation for cognitive training to enhance the cognitive levels of patients. Reuter-Lorenz et al. [68] further proposed that cognitive training might stimulate pre-existing neural reserves or activate neural circuits as "compensatory scaffolds", thereby promoting neural plasticity reorganization to improve MCI.

Exercise is a core component of the management plan for patients with T2DM [69], and its benefits extend beyond metabolic control, significantly enhancing the cognitive function of T2DM patients with MCI. Gallaway et al.'s study [70] also demonstrated that regular physical exercise not only improves cardiovascular and metabolic health but also reduces

the risk of neurodegenerative diseases. This is consistent with the previous meta-analysis conclusions for MCI in the general population [71], jointly confirming the positive role of exercise in improving and maintaining cognitive function. The neuroprotective effect of it is not only due to the direct induction of hippocampal neurogenesis and synaptic plasticity remodeling [72–74], but also various types of exercise therapy (such as aerobic exercise, resistance training, etc.) may exert protective effects on cognitive function through multiple neurobiological mechanisms. At the molecular level, exercise can increase the levels of growth factors in the human body. Among them, brain-derived neurotrophic factor (BDNF), insulin-like growth factor-1 (IGF-1), and vascular endothelial growth factor (VEGF) are currently regarded as the key proteins that are upregulated after exercise, which can promote cell proliferation and growth or the development and function of neurons [75]. Resistance training can promote neural plasticity and improve overall cognitive performance by increasing the concentration levels of IGF-1 in the hippocampus and peripheral blood through activating the neural genesis signaling pathway; aerobic exercise, on the other hand, effectively delays the degenerative changes of the hippocampal structure by increasing the expression of BDNF, thereby improving memory function [76]. In addition, appropriate exercise can also inhibit the release of pro-inflammatory factors and promote the expression of anti-inflammatory factors, thereby reshaping immune homeostasis [77]; at the same time, it can enhance plasma antioxidant activity, alleviate neuronal damage induced by oxidative stress, and thereby synergistically reduce the level of neuroinflammation and achieve multidimensional protection of the nervous system [78].

In the traditional Chinese medicine intervention system for T2DM patients with MCI, the main clinical approaches involve external treatment methods such as massage, acupuncture and moxibustion. Through therapeutic methods such as strengthening the spleen and kidney, tonifying qi and blood, eliminating phlegm and resolving stasis, etc., it aims to delay brain aging and regulate the overall functions of the body to achieve the effects of preventing and treating diseases [79]. In this study, the main intervention measures included in the TCM therapy were acupoint massage and acupuncture therapy. Both of these are important treatment methods of traditional Chinese medicine and have been gradually promoted and applied in clinical practice and have gained recognition from patients. Acupoint massage, grounded in meridian theory, stimulates specific head acupoints to activate meridian energy and blood circulation. Its mechanism of action is to enhance blood circulation in the brain, improve the metabolism of brain cells, thereby promoting the recovery and reconstruction of neural structures related to cognitive functions, improving brain function, slowing down brain atrophy, and thereby enhancing the cognitive function of patients with MCI [80,81].

The principle of acupuncture treatment is similar to that of acupoint massage. Due to its greater stimulation than acupoint massage, it has better therapeutic effects for certain diseases. Yin [82] proposed that the mechanism by which acupuncture improves cognitive function might be related to the regulation of intestinal microbiota and metabolites. There is a bidirectional connection between intestinal microbiota and metabolites and the central nervous system [83], and they are key factors influencing the onset of MCI [84]. Acupuncture can regulate the intestinal microbiota and metabolites, restore the intestinal mucosal barrier function, reduce lipopolysaccharide load and systemic inflammation, and ultimately improve the cognitive function of patients [85]. In this study, there are relatively fewer related studies on TCM therapy, making it impossible to determine which of the two intervention measures has the best effect. Therefore, more research should be conducted in the future to fill this gap.

### 4.3. Limitations

This study has certain limitations: (1) The included literature is scarce, the sample size is small, and there is a lack of long-term follow-up data, which affects the reliability of the results. More large-sample, multi-center, and long-term follow-up high-quality studies are needed to further verify this conclusion in the future. (2) In the quality evaluation section, due to the inherent limitations of the trial design, the difficulty of implementing the concealment of allocation schemes and the blind method for the studies included has increased. Some non-pharmacological intervention measures, such as exercise therapy, cannot blind the researchers and participants. Therefore, only some studies used the blind method and allocation

concealment, which may lead to implementation and measurement biases, resulting in deviations in outcome indicators due to the subjective influence of researchers or participants, and affecting the methodological quality of the included studies. This study strictly formulated the inclusion and exclusion criteria and attempted to control biases as much as possible. (3) There are differences among similar intervention measures. Limited by the number of included studies, detailed subgrouping was not conducted for each type of intervention measure, which might influence the accuracy of the results. (4) This study exhibits considerable heterogeneity. Although we conducted sensitivity analyses and meta-regression on the main outcome indicators, we still failed to identify the source of the heterogeneity.

### 4.4. Implications for clinical practice and future research

This study indicates that non-pharmacological therapies have beneficial effects on the cognitive function of patients with T2DM and MCI. This is consistent with the previous meta-analysis conclusions on non-pharmacological interventions for MCI in the general population [86]. For the top-ranked intervention measures in this study (cognitive training, exercise therapy, traditional Chinese medicine therapy, and comprehensive intervention, etc.), they are also highly in line with existing international guidelines and expert consensus, providing more specific evidence support for their clinical application. The daily management guideline for diabetes-related cognitive impairment recommends that psychologists provide cognitive-based rehabilitation intervention treatment for patients to cope with the decline in cognitive ability [8]. The Chinese Expert Consensus on the Prevention and Treatment of Cognitive Dysfunction in Type 2 Diabetes Mellitus of the Endocrinology Branch of the Chinese Medical Association also recommends using cognitive training therapy for patients with T2DM and MCI, and believes that the combined intervention measures of cognitive training, exercise therapy, and psychological therapy can not only effectively improve the cognitive function of patients, but also be beneficial to the improvement of their daily living ability [87]. Huang et al. [88] believe that aerobic exercise has become a promising intervention measure for MCI, which can simultaneously alleviate the symptoms of T2DM and bring positive feedback to the cognitive function of patients. Traditional Chinese medicine therapy (such as acupoint massage and acupuncture) also shows potential in this study, and its advantages lie in the concepts of "preventing disease before it occurs" and early intervention [89]. Klimova et al. [90] believe that acupuncture and other traditional Chinese medicine therapies can be used as alternative or supplementary treatments to enhance the effectiveness of cognitive function in MCI patients.

Currently, there is an increasing understanding of the cognitive dysfunction associated with diabetes. Compared to drug treatment, non-drug intervention measures are less costly, have fewer adverse reactions, and are more convenient to implement clinically. Therefore, based on the above research findings and existing guideline consensus, in future clinical practice, it is recommended to prioritize the promotion of cognitive training and exercise therapy as basic and effective single intervention measures. When conditions permit, it is recommended to integrate multidisciplinary resources and combine the two for a comprehensive intervention as the preferred choice for cognitive management in patients with T2DM and MCI. In specific clinical practice, healthcare providers should develop personalized intervention strategies and goals based on the patient's health status and physical ability, closely monitor the patient's response during the intervention, and make dynamic adjustments to the intensity, frequency, and content of the intervention. In addition, traditional Chinese medicine hospitals can use acupuncture or acupressure as complementary or alternative treatments to alleviate patient symptoms and prevent further development of cognitive impairment.

Although this study has confirmed the effectiveness of various non-pharmacological therapies, in the traditional Chinese medicine therapy in this study, the effects of acupoint massage and acupuncture therapy have not yet reached a direct comparison conclusion. In the future, more relevant high-quality studies are still needed to further explore the efficacy, patient acceptance, and mechanism of action of the two therapies. In addition, it is suggested that relevant research expand the focus on outcome indicators, conduct comprehensive evaluations of patients' living abilities, mental and behavioral conditions, etc., and adopt more objective indicators to improve the reliability of the results. At the same time, attention should be paid to the long-term effects of the intervention, effectively combine new intervention technologies

such as computers to improve patients' treatment cooperation and compliance, in order to better improve the cognitive function of patients with T2DM and MCI and enhance their quality of life.

## 5. Conclusion

This network Meta-analysis is the first to comprehensively compare the effects of four non-pharmacological intervention measures on the cognitive function of patients with T2DM and MCI. The current evidence indicates that cognitive training, exercise therapy and TCM therapy may be more effective intervention methods for improving the cognitive function of patients with T2DM and MCI, and cognitive training might be the most effective non-pharmacological treatment method. This study holds certain significance for future clinical practice and scientific research. Extensive attention should be given to the impact of cognitive training and exercise therapy on the cognitive function of T2DM patients with MCI, and they should be recommended as supplementary therapeutic modalities for patients. The number of related studies on acupoint massage and acupuncture treatment is relatively small. More large-sample, multi-center randomized controlled trials should be carried out in the future to confirm their effectiveness for patients and reveal their mechanisms and clinical significance.

## Supporting information

**S1 Checklist. PRISMA NMA checklist.**
(DOCX)

**S1 Table. The strategies for database retrieval.**
(PDF)

**S2 Table. The reasons for literature screening exclusion.**
(XLSX)

**S3 Table. ROB 2 risk assessment details.**
(XLSM)

**S4 Table. SUCRA Rank Diagram.**
(DOCX)

**S5 Table. Sensitivity analysis and meta-regression of network meta-analysis.**
(DOCX)

**S1 Fig. The results of sensitivity analysis for the paired meta-analysis.**
(PDF)

**S2 Fig. The pairwise meta-analysis of MoCA.**
(PDF)

**S3 Fig. The pairwise meta-analysis of MMSE.**
(PDF)

**S1 File. Related data for this study.**
(ZIP)

## Acknowledgments

The authors would like to thank all the researchers for providing the data for this work.

## Author contributions

**Data curation:** wenzhuo an, Dongqing Guo, Jie Wang.

**Formal analysis:** wenzhuo an.

**Writing – original draft:** wenzhuo an.

**Writing – review & editing:** Dongqing Guo, Jie Wang, xin chu.

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
