## [Decision Letter · Decision Letter 0]

15 Apr 2025

Dear Dr. chu,

Thank you for submitting your manuscript to PLOS ONE. After careful consideration, we feel that it has merit but does not fully meet PLOS ONE’s publication criteria as it currently stands. Therefore, we invite you to submit a revised version of the manuscript that addresses the points raised during the review process.

We look forward to receiving your revised manuscript.

Kind regards,

Yuzhen Xu

Academic Editor

PLOS ONE

Journal Requirements:

3. As required by our policy on Data Availability, please ensure your manuscript or supplementary information includes the following:

Reviewers' comments:

Reviewer's Responses to Questions

**Comments to the Author**

1. Is the manuscript technically sound, and do the data support the conclusions?

Reviewer #1: Partly

Reviewer #2: Yes

2. Has the statistical analysis been performed appropriately and rigorously?

Reviewer #1: Yes

Reviewer #2: Yes

3. Have the authors made all data underlying the findings in their manuscript fully available?

Reviewer #1: No

Reviewer #2: Yes

4. Is the manuscript presented in an intelligible fashion and written in standard English?

Reviewer #1: Yes

Reviewer #2: Yes

Reviewer #1: Review Comments to the Author

General Comments:

This manuscript explores the impact of non-pharmacological interventions on cognitive function in patients with Type 2 Diabetes Mellitus (T2DM) and mild cognitive impairment (MCI). The study addresses a clinically significant and underexplored topic, and its findings provide valuable insights for both researchers and clinicians. The use of network meta-analysis adds robustness to the methodology, and the inclusion of multiple interventions offers a broad perspective on therapeutic options. However, some areas require improvement to enhance clarity, rigor, and alignment with journal standards.

Major Points:

1. Data Availability:

It is unclear whether all data underlying the findings have been made fully available, as required by PLOS ONE’s data policy. Please ensure that raw data, such as the individual data points behind summary statistics, are provided or deposited in a public repository. If restrictions apply, these should be clearly explained in the manuscript.

2. Results and Clarity:

The results section could benefit from more detailed comparisons of interventions, particularly between cognitive training, exercise therapy, and traditional Chinese medicine therapy. Highlighting specific statistical values in the discussion will improve clarity and support your conclusions.

Consider adding visual aids, such as summary charts or network diagrams, to improve the reader's understanding of key findings.

3.Methodological Transparency:

High heterogeneity was noted in several analyses (e.g., I² > 80%). While random-effects models were used appropriately, a more detailed exploration of potential sources of heterogeneity is necessary.

Some included studies lacked blinding or allocation concealment. Please address how these limitations might affect the reliability of your conclusions and discuss strategies to mitigate these issues in future research.

4.Language and Readability:

While the manuscript is written in standard English, certain sections, particularly the discussion, contain overly complex sentences. Simplifying these for better readability would enhance the manuscript.

Ensure consistent terminology throughout the manuscript (e.g., “traditional Chinese medicine therapy” vs. “TCM therapies”).

5.Figures and Tables:

Improve the legends for figures and tables to provide more detailed explanations of the data presented. Consider using comparative charts to summarize the effectiveness of interventions (e.g., SUCRA rankings).

Minor Points:

1. Formatting and Compliance:

Adhere strictly to PLOS ONE’s formatting guidelines, including consistent citation formatting and a clear structure for supplementary materials.

2. Discussion:

Expand the discussion on the practical applications of your findings and suggest specific directions for future research, such as evaluating long-term effects or combining interventions.

Conclusion:

This manuscript has the potential to make a significant contribution to the field, but revisions are necessary to address data availability, methodological transparency, and clarity in the results presentation. These improvements will strengthen the manuscript’s scientific rigor and impact.

Reviewer #2: I reviewed this article and found it insightful. However, I noticed that while the references mention "Traditional Chinese Medicine" (TCM), the content primarily focuses on acupuncture.

Since TCM is a broad field encompassing various practices such as herbal medicine, cupping, moxibustion, and dietary therapy and Acupuncture, it might be more accurate and clear to replace "Traditional Chinese Medicine" with "Acupuncture" throughout the article. This adjustment would help ensure consistency and better reflect the article’s actual focus.

Thank you for your contribution, and I appreciate your consideration of this suggestion.

**Do you want your identity to be public for this peer review?** For information about this choice, including consent withdrawal, please see our Privacy Policy

Reviewer #1: **Yes: ** Dr. Akram Aldilaimi

Reviewer #2: **Yes: ** Dr. Momtaz Ahmed

---

## [Author Response · Author response to Decision Letter 1]

25 Apr 2025

Dear editors and reviewers,

We are very grateful for your constructive comments and suggestions for our manuscripts entitled “Effects of non-pharmacological interventions on cognitive function in patients with type 2 diabetes mellitus and mild cognitive impairment: a network Meta-analysis” (ID: PONE-D-24-54429). Your comments are very valuable and helpful for improving our manuscripts.in the following, the responses to all the comment are provided one by one.

We have tried our best to make all the revisions clear, and we hope that the revised manuscript can satisfy the requirements for publication.

The main revisions in the new manuscript are:

Reply to the editor’s suggestions for article modification

1. This study was modified according to the format template for viewing PLOS ONE, which meets the format requirements of PLOS ONE.

2. We have registered an ORCID iD: 0009-0004-4777-6524.

3. We have explained in the main text and Supplementary Information S2 Table the reasons for excluding each study after reading the full text. S2 Table also provides the first author, year and title of each excluded study. All the included and excluded literatures do not include unpublished studies.

4. We included the name of the literature manager and the date of data extraction. We extracted the data using EndNote and placed the extracted data in the Supplementary Information, S1 File. And it provided the main result data of paired Meta-analysis, as well as the league table of network Meta-analysis and the SUCRA values of each intervention measure.

5. We supplemented a complete risk assessment table for each item included in the study, which was made using Revman and placed in Fig 2 and Supplementary Information S1 Fig.

Reply to the reviewer's suggestions for article modification

1. Clarity in Results Section: In the manuscript, we have described the quantity of each intervention measure. In the abstract, results and discussion sections, we have highlighted the key statistical values to enhance the clarity of the results.

2. Depth of Analysis: Due to the abundance of content in the discussion, we have added subheadings to enhance clarity. We have delved deeply into the neurobiological mechanisms by which exercise therapy and traditional Chinese medicine can improve the cognitive function of patients with T2DM and MCI. In the exercise therapy, we have provided an in-depth description of the mechanisms of action of relevant growth factors and inflammatory cytokines. Since the main traditional Chinese medicine therapies included in this study are acupressure and acupuncture, we have explored the mechanisms of these two therapies. See page 21-23 of the manuscript for details.

3. Addressing Limitations: We conducted a thorough analysis of the limitations of the methodology, and discussed the reasons and countermeasures for the biases caused by the few included literatures, the lack of blinding and allocation concealment in some studies. See page 23 of the manuscript for details.

4. Consistency and Terminology: We have read through the entire manuscript and unified the key terms. For instance, we will define the traditional Chinese medical therapy that appears for the first time in the text as TCM therapy, and henceforth it will be referred to as such throughout the text. We also simplified the more complex sentences in the original discussion section, especially in the part that describes the mechanism of action of acupoint massage in traditional Chinese medicine. Some professional terms used in traditional Chinese medicine might be rather obscure and difficult to understand. We optimized them to make the manuscript more accessible to a wider range of readers. See page 22 of the manuscript for details.

5. Future Directions: We have delineated the directions for future clinical practice and research in this field, including standardizing operation procedures, conducting long-term follow-ups, comprehensively evaluating outcome indicators, and adopting new technologies to enhance patient compliance. Additionally, we have proposed that more high-quality research should be conducted in the future to further explore the efficacy, patient acceptance, and mechanism of action between acupoint massage and acupuncture. See page 24 of the manuscript for details.

Reply to points

1. Data Availability: All relevant data are within the manuscript and its appendix files.

2. Results and Clarity: We have respectively provided explanations in the text regarding the number of included literature for each intervention measure in each outcome indicator. Specific statistical values have been highlighted in the abstract, results and discussion sections of the manuscript.

3. Methodological Transparency: Regarding the high heterogeneity observed in the pairwise Meta-analysis results, we conducted sensitivity analyses for each intervention measure in each of the two outcome indicators separately. The results indicated that: In the comprehensive intervention where MoCA score was used as the outcome indicator, after eliminating the study by Ye et al., the heterogeneity decreased. The analysis suggests that this might be due to the fact that in the comprehensive intervention measures, the cognitive training component of this study accounted for a relatively large proportion while other intervention measures were relatively less prominent. In the cognitive training with MMSE score as the outcome indicator, after eliminating the study by Liang et al., the heterogeneity decreased. The analysis might be attributed to the relatively short duration of the intervention in this study. There was no significant change in the heterogeneity of other intervention measures. The Meta-analysis results were robust. The results of the sensitivity analysis can be found in Supplementary Information S2 Fig. The heterogeneity of the network meta-analysis is relatively small.

We conducted a thorough analysis of the limitations of the methodology, and discussed the reasons and countermeasures for the biases caused by the few included literatures, the lack of blinding and allocation concealment in some studies. See page 23 of the manuscript for details.

4. Language and Readability: We have made partial revisions to overly complex sentences and checked the entire manuscript to ensure consistency in terminology.

5. Figures and Tables: We plotted cumulative probability ranking diagrams (Fig 4 and Fig 5) for the two outcome indicators of different intervention measures, and sorted the intervention measures based on the SUCRA values. The detailed tables can be found in Supplementary Information S3 Table.

6. Formatting and Compliance: This study strictly adhered to the format guidelines of PLOS ONE and revised the citation format and supplementary information.

7. Discussion: We have expanded the discussion section, pointed out the directions for future clinical practice and scientific research, and proposed that personalized comprehensive intervention measures should be carried out based on the specific conditions of patients and long-term follow-up should be conducted to verify the effectiveness and safety of the measures, with the aim of better improving the cognitive function of patients with T2DM and MCI and enhancing their quality of life. For details, please refer to page 24 of the manuscript.

Sincerely,

Corresponding author.

---

## [Decision Letter · Decision Letter 1]

25 Jun 2025

Dear Dr. Chu,

Thank you for submitting your manuscript to PLOS ONE. After careful consideration, we feel that it has merit but does not fully meet PLOS ONE’s publication criteria as it currently stands. Therefore, we invite you to submit a revised version of the manuscript that addresses the points raised during the review process.

We look forward to receiving your revised manuscript.

Kind regards,

Yuzhen Xu

Academic Editor

PLOS ONE

Journal Requirements:

Reviewers' comments:

Reviewer's Responses to Questions

**Comments to the Author**

Reviewer #3: All comments have been addressed

Reviewer #4: All comments have been addressed

2. Is the manuscript technically sound, and do the data support the conclusions?

Reviewer #3: Yes

Reviewer #4: Yes

3. Has the statistical analysis been performed appropriately and rigorously?

Reviewer #3: Yes

Reviewer #4: Yes

4. Have the authors made all data underlying the findings in their manuscript fully available?

Reviewer #3: Yes

Reviewer #4: Yes

5. Is the manuscript presented in an intelligible fashion and written in standard English?

Reviewer #3: Yes

Reviewer #4: Yes

Reviewer #3: This manuscript presents a well-conducted network meta-analysis comparing non-pharmacological interventions for cognitive impairment in patients with type 2 diabetes and MCI. The study design is rigorous, the literature search comprehensive, and the statistical methods are appropriate. The findings—particularly the SUCRA rankings—offer valuable clinical insights. However, several methodological and reporting details require clarification and enhancement before this work is suitable for publication.

Major Suggestions :

1.Methods Sections 1.1–1.2 (Literature Search and Eligibility Criteria):

Provide detailed database search information (PubMed, Web of Science, Embase, Cochrane Library, CNKI, VIP, Wanfang, SionMed), including search dates and full search strings in the main text or appendix. Clearly list each inclusion and exclusion criterion (e.g., MCI diagnostic criteria, intervention definitions, outcome measures) and include a PRISMA flow diagram to ensure reproducibility.

2.Methods Section 1.4 (Quality Assessment):

Specify the version of the risk‑of‑bias tool used (e.g., Cochrane RoB 2.0) and describe the criteria for judging each domain (random sequence generation, allocation concealment, blinding, data completeness, selective reporting, etc.). Present a summary table of risk‑of‑bias judgments (low/unclear/high) for all included studies rather than only a figure.

3.Methods Section 1.5 (Statistical Analysis):

Justify the choice of fixed‑effect versus random‑effect models (e.g., use random‑effect when I² ≥ 50%). Detail the methods for testing consistency (e.g., node‑splitting or design‑by‑treatment interaction models) and assessing heterogeneity (I², τ²). Specify the software and package versions (e.g., R version, Stata version, relevant R packages/functions).

4.Results Sections 2.4 (Pairwise Meta‑Analysis) & 2.5.2.3 (Network Meta‑Analysis Convergence and Sensitivity):

Incorporate sensitivity analyses for both pairwise and network meta‑analyses. Show forest plots or tables comparing heterogeneity statistics (I²) and effect estimates before and after excluding high‑risk studies or switching between fixed and random effects, and discuss the impact on conclusions.

5.Results Section 2.5 (Sources of Heterogeneity and Subgroup/Meta‑Regression Analyses):

For comparisons with high I² (>50%), perform subgroup analyses or meta‑regression by intervention type (cognitive training, exercise, TCM, combined interventions), intervention duration (≤3 months vs. >3 months), baseline cognitive level, or patient age to explore potential moderators of effect.

6.Results Section 2.5.2.2 (SUCRA Ranking Interpretation):

In the Results or Discussion, explain the meaning of SUCRA (0–100% scale, with higher values indicating a higher probability of being the best intervention) and relate these rankings to MD values and 95% CIs to guide interpretation of each intervention’s relative efficacy.

7.Discussion Section 3 (Clinical Implications):

Deepen the discussion of clinical relevance by comparing top‑ranked interventions (e.g., combined, cognitive training, exercise) with existing clinical guidelines or consensus statements for cognitive care in T2DM patients, highlighting which interventions are currently recommended or readily implementable, and offering practice recommendations based on your findings.

8.Expand Discussion on Mechanisms and Clinical Implications:

Elaborate on the potential neurobiological mechanisms by which cognitive training, exercise, and TCM may improve cognition in T2DM + MCI patients. Discuss practical considerations for implementing these interventions in clinical settings and compare your results with existing meta-analyses in general MCI populations to highlight novel contributions.

Reviewer #4: The findings highlight cognitive training and exercise therapy as the most effective non-pharmacological interventions, offering clinicians evidence-based alternatives to pharmacotherapy for T2DM patients with MCI.

**Do you want your identity to be public for this peer review?** For information about this choice, including consent withdrawal, please see our Privacy Policy

Reviewer #3: No

Reviewer #4: **Yes: ** KUO TING TANG

---

## [Author Response · Author response to Decision Letter 2]

2 Jul 2025

Dear editors and reviewers,

We are very grateful for your constructive comments and suggestions for our manuscripts entitled “Effects of non-pharmacological interventions on cognitive function in patients with type 2 diabetes mellitus and mild cognitive impairment: a network Meta-analysis” (ID: PONE-D-24-54429). Your comments are very valuable and helpful for improving our manuscripts.in the following, the responses to all the comment are provided one by one.

We have tried our best to make all the revisions clear, and we hope that the revised manuscript can satisfy the requirements for publication.

Reply to the editor’s suggestions for article modification

1.We reviewed the reference list of this article to ensure that no citations were made to withdrawn papers. All the references have been revised in accordance with the formatting requirements of the PLOS ONE journal.

2.All the graphic files have been modified using the PACE digital diagnostic tool.

Reply to the reviewer's suggestions for article modification

1.“Methods Sections 1.1–1.2 (Literature Search and Eligibility Criteria):

Provide detailed database search information (PubMed, Web of Science, Embase, Cochrane Library, CNKI, VIP, Wanfang, SionMed), including search dates and full search strings in the main text or appendix. Clearly list each inclusion and exclusion criterion (e.g., MCI diagnostic criteria, intervention definitions, outcome measures) and include a PRISMA flow diagram to ensure reproducibility. ”

Response: In the article, we have specified the specific databases and the retrieval dates for the literature search, and provided detailed retrieval information for each database in Supplementary Information S1 Table. We clearly listed each inclusion and exclusion criterion, revised the MCI diagnostic criteria (please refer to page 7 of this manuscript), and clearly defined and cited relevant references based on the American College of Sports Medicine and previous studies regarding the specific definitions of various non-pharmacological intervention measures (please refer to page 7 of this manuscript). Regarding the outcome indicators, we further clarified the outcome indicators of this study and pointed out that the higher the scores of MoCA and MMSE tests, the better the cognitive function of the patients (please refer to page 8 of this manuscript). And we strictly followed the PRISMA guidelines to re-create the literature screening flowchart.

2. “Methods Section 1.4 (Quality Assessment):

Specify the version of the risk of bias tool used (e.g., Cochrane RoB 2.0) and describe the criteria for judging each domain (random sequence generation, allocation concealment, blinding, data completeness, selective reporting, etc.). Present a summary table of risk of bias judgments (low/unclear/high) for all included studies rather than only a figure.”

Response: We specified the version of the bias risk assessment tool used (RoB 2.0) and introduced six aspects for evaluating the quality of the literature. Each dimension was classified as "yes" (low risk), "no" (high risk), or "unclear" (moderate risk). We re-generated the quality evaluation chart for the included literature (Fig 2) based on RoB 2.0. The summary table for the assessment of bias risk for all included studies is presented in Supporting information S3 Table.

3. “Methods Section 1.5 (Statistical Analysis):

Justify the choice of fixed‑effect versus random‑effect models (e.g., use random‑effect when I² ≥ 50%). Detail the methods for testing consistency (e.g., node‑splitting or design‑by‑treatment interaction models) and assessing heterogeneity (I², τ²). Specify the software and package versions (e.g., R version, Stata version, relevant R packages/functions).”

Response: We specified the versions of the software and packages used for data analysis (Stata/SE 15.1 and R 4.3.1, with data analysis conducted through the gemtc package), and explained that the effect model was selected based on the size of heterogeneity I2. When I2 was less than 50%, a fixed effect model was chosen; otherwise, a random effect model was used. If a closed loop was formed, the method of node splitting was employed for the inconsistency test.( please refer to page 9-10 of this manuscript)

4. “Results Sections 2.4 (Pairwise Meta‑Analysis) & 2.5.2.3 (Network Meta‑Analysis Convergence and Sensitivity):

Incorporate sensitivity analyses for both pairwise and network meta‑analyses. Show forest plots or tables comparing heterogeneity statistics (I²) and effect estimates before and after excluding high‑risk studies or switching between fixed and random effects, and discuss the impact on conclusions.”

Response: Our previous study had already conducted a sensitivity analysis for the pairwise Meta-analysis. The detailed results can be found in Supporting information S1 Fig. According to your suggestion, we conducted heterogeneity tests for the two outcome indicators in the network Meta-analysis and presented their respective I2, Dbar, pD and DIC values under the fixed effect model and the random effect model (on pages 16 and 18 of the manuscript). Due to the influence of the number of included studies, we conducted a sensitivity analysis for the included studies with MoCA score as the outcome indicator (on pages 19-20 of the manuscript). Unfortunately, we were unable to identify the source of heterogeneity. By excluding the study with the smallest sample size for the sensitivity analysis, we observed that the ranking of SUCRA did not change, indicating that our conclusion is relatively reliable. We have placed the results of the sensitivity analysis in Supporting information S5 Table, including the comparison of heterogeneity analysis data before and after the sensitivity analysis.

5. “Results Section 2.5 (Sources of Heterogeneity and Subgroup/Meta‑Regression Analyses):

For comparisons with high I² (>50%), perform subgroup analyses or meta‑regression by intervention type (cognitive training, exercise, TCM, combined interventions), intervention duration (≤3 months vs. >3 months), baseline cognitive level, or patient age to explore potential moderators of effect.”

Response: We conducted a Meta-regression using the outcome indicators of the MoCA score in the included studies. We analyzed the data by considering whether the intervention duration was > 3 months and whether the intervention frequency was > 3 times per week as covariates. However, we still failed to identify the source of heterogeneity. Therefore, we have to admit that the large heterogeneity is one of the limitations of this study. Detailed information can be found in Supporting information S5 table.

6. “Results Section 2.5.2.2 (SUCRA Ranking Interpretation):

In the Results or Discussion, explain the meaning of SUCRA (0–100% scale, with higher values indicating a higher probability of being the best intervention) and relate these rankings to MD values and 95% CIs to guide interpretation of each intervention’s relative efficacy.”

Response: In the 1.5 statistical methods section in this article, we explained the meaning of SUCRA, indicated that the range of SUCRA values is from 0 to 1, and stated that the larger the value, the better the effect of the intervention measure. We also provided explanations in the SUCRA ranking results of the two outcome indicators. At the same time, in the 1.5 statistical methods section, we explained the meaning of the MD value, stating that when the MD value is ≥ 0.8, it indicates a large effect size. In the results of the network Meta-analysis of the two outcome indicators, we respectively explained the MD values of the intervention measures. In the 3.1 main results section of the discussion part of this article, we combined SUCRA and MD values to explain the reliability of cognitive training as the best intervention measure for improving patients' cognitive function.

7. “Discussion Section 3 (Clinical Implications):

Deepen the discussion of clinical relevance by comparing top‑ranked interventions (e.g., combined, cognitive training, exercise) with existing clinical guidelines or consensus statements for cognitive care in T2DM patients, highlighting which interventions are currently recommended or readily implementable, and offering practice recommendations based on your findings.”

Response: Based on your suggestions, we extensively reviewed international guidelines and expert consensus, and combined existing clinical guidelines and expert opinions to explain several top-ranked non-pharmacological intervention measures (please refer to page 27 of this manuscript). Based on the above findings, we recommend that in clinical practice, cognitive training and exercise therapy be prioritized as basic and effective single intervention measures. When conditions permit, a combined comprehensive intervention combining the two should be given priority. Traditional Chinese medicine hospitals can also use traditional Chinese medicine treatments as supplementary or alternative therapies. At the same time, we have provided some specific suggestions for future clinical work for reference (please refer to page 28 of this manuscript).

8. “Expand Discussion on Mechanisms and Clinical Implications:

Elaborate on the potential neurobiological mechanisms by which cognitive training, exercise, and TCM may improve cognition in T2DM + MCI patients. Discuss practical considerations for implementing these interventions in clinical settings and compare your results with existing meta-analyses in general MCI populations to highlight novel contributions.”

Response: Based on your suggestions, we reorganized and elaborated in depth the potential neurobiological mechanisms by which cognitive training, exercise therapy and traditional Chinese medicine might improve the cognition of patients with T2DM and MCI. We discussed our ideas in clinical practice and compared our research with a previous meta-analysis on the cognitive function of MCI patients in the general population. The results were basically consistent. (See pages 22-25 and 28 of this manuscript.)

Sincerely,

Corresponding author.

---

## [Decision Letter · Decision Letter 2]

16 Jul 2025

Effects of non-pharmacological interventions on cognitive function in patients with type 2 diabetes mellitus and mild cognitive impairment: a network Meta-analysis

PONE-D-24-54429R2

Dear Dr. Chu,

We’re pleased to inform you that your manuscript has been judged scientifically suitable for publication and will be formally accepted for publication once it meets all outstanding technical requirements.

Kind regards,

Yuzhen Xu

Academic Editor

PLOS ONE

Additional Editor Comments (optional):

Reviewers' comments:

Reviewer's Responses to Questions

**Comments to the Author**

Reviewer #3: All comments have been addressed

Reviewer #4: All comments have been addressed

2. Is the manuscript technically sound, and do the data support the conclusions?

Reviewer #3: Yes

Reviewer #4: Partly

3. Has the statistical analysis been performed appropriately and rigorously?

Reviewer #3: Yes

Reviewer #4: Yes

4. Have the authors made all data underlying the findings in their manuscript fully available?

Reviewer #3: Yes

Reviewer #4: Yes

5. Is the manuscript presented in an intelligible fashion and written in standard English?

Reviewer #3: Yes

Reviewer #4: Yes

Reviewer #3: (No Response)

Reviewer #4: (No Response)

**Do you want your identity to be public for this peer review?** For information about this choice, including consent withdrawal, please see our Privacy Policy

Reviewer #3: No

Reviewer #4: No
